# Comparison of 7 surgical interventions for recurrent lumbar disc herniation: A network meta-analysis and systematic review

Hang Zhang[1], Junmao Gao[2]*, Qipeng Xie[3◉], Mingxin Zhang[1◉]

1 College of Integrated Chinese and Western Medicine, Hebei University of Chinese Medicine, Shijiazhuang, Hebei Province, China, 2 Department of Orthopaedics, Hebei University of Chinese Medicine affiliated Yiling Hospital, Shijiazhuang, Hebei Province, China, 3 Department of Orthopaedics, Changning Hospital of Traditional Chinese Medicine, Changning, Sichuan Province, China

◉ These authors contributed equally to this work.
* 2386268591@qq.com

**Data Availability Statement:** All relevant data are within the paper and its Supporting information files.

## Abstract

### Study design

Network meta-analysis of multiple treatment comparisons of recurrence lumbar disc herniation.

### Objective

The purpose of comparing the differences between different surgical approaches for recurrent lumbar disc herniation (LDH).

### Methods

The PubMed, Embase, MEDLINE, Cochrane Library, Web of Science, Google Scholar and China National Knowledge Infrastructure databases were searched for articles published before April 10th, 2024. The Markov chain Monte Carlo methods were used to perform a hierarchical Bayesian NMA in R version 4.3.3 using a random effects consistency model. The assessing outcomes were pain intensity, disability, complications and recurrence.

### Results

20 studies including 1556 patients and 7 different approaches (PELD, MED, MIS-TLIF, TLIF, Unilat -TLIF, PLIF and OD) were retrospectively retrieved. the efficacy of each approach was the same in relieving pain, OD was significantly better than PELD and MIS-TLIF in relieving dysfunction (SMD: 1.9[0.21,3.4] and 2.0[0.084,3.8], respectively), In addition, MIS-TLIF was significantly lower than PELD and MED in the complication rate (SMD: 0.37[0.14,0.84] and 0.15[0.034,0.68], respectively), TLIF was significantly Lower than MED in the complication rate (SMD:0.14 [0.027,0.70]), PELD was significantly higher than MIS-TLIF, TLIF and PLIF in the recurrence rate (SMD: 1.3e-17 [2.4e-44,0.00016],1.2e-12 [2.1e-36,0.34] and 1.4e-12[6.2e-35,0.013], respectively), MED was significantly higher than MIS-TLIF and PLIF in the recurrence rate (SMD: 2.6e-17[5.6e-44,0.0037] and

**Funding:** The author(s) received no specific funding for this work.

**Competing interests:** The authors have declared that no competing interests exist.

**Abbreviations:** MED, Microendoscopic Discectomy; MIS-TLIF, Minimally Invasive transforaminal lumbar interbody fusion; OD, Open discectomy; ODI, Oswestry disability index; OR, Odds ratios; PELD, Percutaneous Endoscopic Lumbar Diskectomy; PLIF, posterior lumbar interbody fusion; RCT, randomized controlled trial; RLDH, Recurrent Lumbar Disc Herniation; SMD, standardized mean differences; TLIF, transforaminal lumbar interbody fusion; Unilat TLIF, Unilateral transforaminal lumbar interbody fusion; VAS, visual analogue scale.

3.1e-12[1.6e-34,0.022], respectively), OD was significantly higher than MIS-TLIF, TLIF and PLIF in the recurrence rate (SMD:4.6e+16[2.3e+02,3.0e+43], 4.3e+11[2.4,2.5e+35] and 4.1e+11[35,8.7e+33], respectively).

## Conclusions

In the treatment of recurrent lumbar disc herniation, vertebral fusion surgery is superior to repeat discectomy. At the same time, MIS-TLIF may be a preferable surgical procedure in the treatment of recurrent lumbar disc herniation.

## Introduction

The efficacy and safety of surgical intervention for recurrent lumbar disc herniation (RLDH) have gained widespread acceptance in clinical practice [1–3]. The recurrence of nucleus pulposus extrusion at the same level and degeneration of the intervertebral disc are the primary etiological factors contributing to RLHD [4], with reported incidence ranging from 2% to 25%. Currently, reoperation for recurrent lumbar disc herniation is still considered one of the treatment options [5–7]. The surgical interventions for RLDH include Open discectomy (OD), Microendoscopic Discectomy (MED), Minimally Invasive transforaminal lumbar interbody fusion (MIS-TLIF), Percutaneous Endoscopic Lumbar Diskectomy (PELD), Posterior Lumbar Interbody Fusion (PLIF), Transforaminal Lumbar Interbody Fusion (TLIF), Unilateral transforaminal lumbar interbody fusion (Unilat TLIF). However, there is still controversy regarding whether vertebral fusion is necessary after reoperation for recurrent lumbar disc herniation [2]. In clinical practice, most surgeons advocate for vertebral fusion after reoperation for recurrent lumbar disc herniation. This is mainly because the scope of damage to the articular surfaces is greater after reoperation compared to the initial lumbar disc reoperation, which further exacerbates the risk of lumbar instability [8, 9]. The latest guidelines indicate that in the case of recurrent lumbar disc herniation with lumbar instability or chronic refractory low back pain, fusion is recommended [2]. Therefore, when patients do not present indications for such surgery, surgeons face a certain challenge in deciding on the surgical approach. Although Feng Lei et al. [10]. believe that vertebral fusion is superior to repeated disc surgery in the treatment of recurrent lumbar disc herniation, they only conducted a controlled study comparing vertebral fusion and repeated disc surgery and did not investigate the differences among different surgical methods. In order to enhance the feasibility of new evidence, we conducted this network meta-analysis study to demonstrate whether vertebral fusion is better than repeated lumbar disc reoperation in the treatment of recurrent lumbar disc herniation. Additionally, we compared the differences among different surgical methods through direct and indirect comparisons.

## Materials and methods

### Literature search strategy

PubMed, Embase, MEDLINE, Cochrane Library, Web of Science, Google Scholar and China National Knowledge Infrastructure databases were searched for articles published before April 10th, 2024, with the search terms of (relapse OR recurrent OR repeat OR recurrence or reoperation) AND (lumbar OR Back OR lumbosacral region OR Lumbar vertebra OR Lumbar spine) AND (Intervertebral Disc Displacement OR Intervertebral Disc Degeneration OR

Protruded Disc OR Protruded Disk OR Intervertebral Disk Displacement OR Intervertebral Disk Herniation OR Slipped Disk OR Slipped Disc OR Disk Prolapse OR Prolapsed Disk OR Disk Prolapse OR Herniated Disc OR Herniated Disk OR Slipped Disc OR Prolapsed Disc OR Disc Herniation OR Intervertebral Disc Herniation OR Intervertebral Disk Herniation OR Disk Herniation OR Disk Herniated OR Intervertebral Disk Protrusion OR Intervertebral Disc Protrusion OR Disc Protrusion OR Disk Protrusion OR Protruded Disc) of PubMed. Duplicate articles were excluded, titles and abstracts of search results were screened for preliminary eligibility, and retrieved full-texts were evaluated by 2 independent reviewers. Any disagreements were solved through consensus with a third researcher. We also chose references cited in the articles and relevant review articles to identify additional studies. In addition, this study was registered in the international register of systematic reviews (Prospero registration number: CRD42024545798).

### Inclusion criteria for studies

For this systematic review, our inclusion criteria were the following:

1. Studies on surgery for RLDH, which must be confirmed by magnetic resonance imaging;

2. Measured one of the clinical outcomes (i.e. visual analog scale (VAS)for leg or back pain, Oswestry Disability Index (ODI) [11]), operative variables (i.e. operative time, blood loss, length of stay), and complications (i.e. neurological deficit, dural tear, segmental instability, re-recurrence, reoperations, surgical site, and other infections);

3. Prospective and retrospective studies with follow-up longer than 12 months.

### Exclusion criteria for studies

We excluded studies with re surgery details not described in the study, cadaveric studies, animal studies, case reports, biomechanical studies, review articles, letters, editorials, abstracts, interim reports, and comments. There was no language restriction on study eligibility.

### Data extraction

From eligible studies, extracted data included authors, year of publication, country, methodology, treatments, demographic, clinical outcomes (i.e. VAS for leg and back pain, ODI), complications (i.e. neurological deficit, dural tear, segmental instability, recurrence, reoperations, surgical site, and other infections) and operative variables (length of hospital stay, blood loss, operative time). If data were available at multiple time points within the reporting window, we extracted data at the longest follow-up period. The missing data were retrieved by contacting the authors. Additionally, we adhered to the Cochrane Review Handbook's instructions for data diversion.

### Assessment of risk of bias

Two review authors independently assessed the methodological quality of articles through the Methodological Index for Non-Randomized Studies (MINORS) score (Table 1). MINORS [12] is a validated score with 12 items for comparative studies and 8 items for case series which explored the aim of the study, patient inclusion criteria, collection of data, endpoints, follow-up and rate of loss at follow-up, calculation of study sample, presence of a control group, equivalence between groups and an adequate statistics. Each category has a score of 0 if the datum is not reported, 1 if it is partially reported, and 2 if it is well established. The best

**Table 1. Methodological index for non-randomized studies, MINORS.**

| | | | |
|---|---|---|---|
| A. A clearly stated aim | NR | PR | WE |
| B. Inclusion of consecutive patients | NR | PR | WE |
| C. Prospective collection of data | NR | PR | WE |
| D. Endpoints appropriate to the aim of the study | NR | PR | WE |
| E. Unbiased assessment of the study endpoint | NR | PR | WE |
| F. Follow-up period appropriate to the aim of the study | NR | PR | WE |
| G. Loss to follow-up less than 5% | NR | PR | WE |
| H. Prospective calculation of the study size | NR | PR | WE |

NR: not reported (0 point); PR: partially reported (1 point); WE: well established (2 point)

A: the question addressed should be precise and relevant in the light of available literature.

B: all patients potentially fit for inclusion (satisfying the criteria for inclusion) have been included in the study during the study period (no exclusion or details about the reasons for exclusion).

C: data were collected according to a protocol established before the beginning of the study.

D: unambiguous explanation of the criteria used to evaluate the main outcome which should be in accordance with the question addressed by the study. Also, the endpoints should be assessed on anintention-to-treat basis.

E: blind evaluation of objective endpoints and double-blind evaluation of subjective endpoints. Otherwise, the reasons for not blinding should be stated.

F: the follow-up should be sufficiently long to allow the assessment of the main endpoint and possible adverse events.

G: all patients should be included in the follow-up. Otherwise, the proportion lost to follow-up should not exceed the proportion experiencing the major endpoint.

H: information of the size of detectable difference of interest with a calculation of 95% confidence interval, according to the expected incidence of the outcome event, and information about the level for statistical significance and estimates of power when comparing the outcomes.

methodological quality of the comparative paper is set at 24 points, while the case series can reach a maximum of 16 points. The Risk of Bias in Non-Randomized Studies of Interventions (ROBINS-I) tool [13] was used to assess the risk of bias for included studies in this paper. The risk of bias of 2 RCTs was evaluated using the Cochrane Collaboration tool [14]. Any disagreement during the process of data extraction and quality assessment would be solved by discussion with the third author.

## Statistical analysis

The raw data required for this study are listed in S5 Table. Odds ratios (ORs)were estimated for dichotomous outcomes, and standardized mean differences (SMD) were estimated for continuous outcomes. Because of the heterogeneity between studies, a random effects model was used for NMA. Forest plots and the $I^2$ statistic were used to investigate heterogeneity. Heterogeneity between different studies was evaluated by $I^2$, and $P < 0.05$ was considered statistically significant. $I^2$ values of $< 25\%$, $25\%$ to $75\%$, and $> 75\%$ represented mild, moderate, and severe heterogeneity, respectively [15]. We compared NMA results (indirect results) with pairwise meta-analysis results (direct results) to explore the causes of inconsistencies. The network geometry of NMA was performed using statistical analysis software Stata version 17.0. The Markov chain Monte Carlo methods were used to perform a hierarchical Bayesian NMA in R version 4.3.3 [16] using a random effects consistency model [17–19]. The estimated result of each relative treatment effect was a combination of direct evidence between the different treatments and indirect evidence from an NMA. We assumed that they were consistent. When there was no direct connection between the different treatments, the effect estimate could only

come from indirect evidence [18, 20]. We used the noninformative prior distribution and the overdispersed initial value in the models of the 4 chains to fit the model, yielding 80,000 iterations, and the refinement interval of each chain was 10 times. To rank the treatments, we used 2 ways. Firstly, we used posterior probabilities of outcomes to calculate probabilities of treatment ranking. Secondly, we used the surface under the cumulative ranking probabilities (SUCRA) to indicate which treatment was the best one [21]. Inconsistency was evaluated by comparing statistics for the deviance information criteria in fitted consistency and inconsistency models and by node-split; P < 0.05 suggested significant inconsistency [22].

## Results

### Systematic review and qualitative assessment

The flow chart of the study screening process and the main reasons for elimination are shown in Fig 1. We identified 7697 studies, of which 20 studies were included (with data for 1556 participants) and 7 approaches (PELD, MED, MIS-TLIF, TLIF, Unilat -TLIF, PLIF and OD) were included (Fig 2). The 20 studies were published between 2013 and 2024. There were 2 RCTs and the remaining were retrospective studies. The summary of study characteristics is presented in Table 2 [23–42]. The Methodological Index for Non-Randomized Studies (MINORS) tool was used to assess the retrospective studies included in this study (Table 3). Overall, the methodological quality score of eighteen retrospective studies varied from 12 to 15. Areas of significant concern were items 5 (prospective collection of data) and 8(Prospective calculation of the study size). The risk of retrospective studies bias (RoB) was assessed using the ROBINS-I tool. High RoB was found in 16/18 articles, and Intermediate RoB was found in 2/18 articles (Fig 3). Domains of most concerns across studies were biased due to the selection of participants, and bias in the measurement of outcomes. In addition,2 RCTs were assessed using the Cochrane Collaboration tool. The risk of bias for 2 RCTs was assessed as high. Areas of significant concern were blind of outcome assessment (Fig 4).

### Pain

Sixteen studies (80%) with 1456 patients (92.98%) presented usable results for VAS (back pain) (7 approaches). In addition, five studies (25%) with 616 patients (39.59%) presented usable results for VAS (leg pain) (3 approaches). In the consistency model, There was no significant difference in the improvement of VAS (back pain) or VAS (leg pain) between any 2 different approaches (Fig 5A), In VAS (back pain) or VAS (leg pain), the results obtained in the consistency model were in good agreement with those obtained in the inconsistency model, and there is no significant difference in node-splitting analysis (all P > 0.05; Fig 6; S1 Table). The direct and indirect results of different approaches are shown in Fig 6. These results indicate that the efficacy profile of each approach was the same for VAS (back or leg pain). The cumulative ranking probabilities (SUCRA) of VAS (back or leg pain) change ranking from high to low is shown in Fig 5B. The probabilities are detailed in the S2 Table.

### ODI

Eleven studies (55%) with 1191 patients (76.54%) presented usable results for ODI (6 approaches). In the consistency model, OD was significantly better than PELD and MIS-TLIF in relieving patients' ODI (SMD: 1.9[0.21,3.4] and 2.0[0.084,3.8], respectively) (Fig 7A). The results obtained in the consistency model were in good agreement with those obtained in the inconsistency model, and the results obtained in the consistency model were in good agreement with those obtained in the inconsistency model, and there is no significant difference in

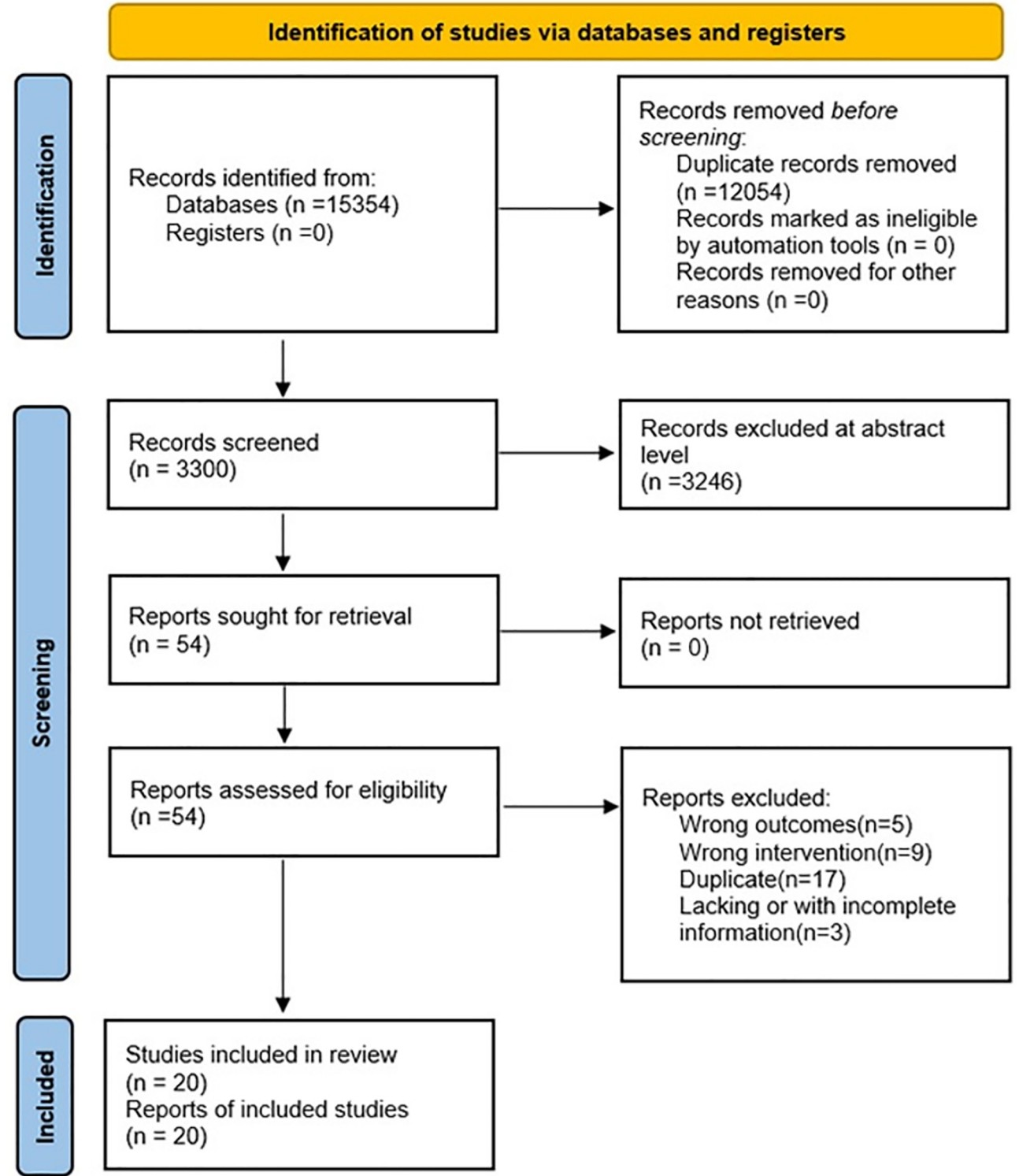

**Fig 1. Flowchart of study selection and design.**

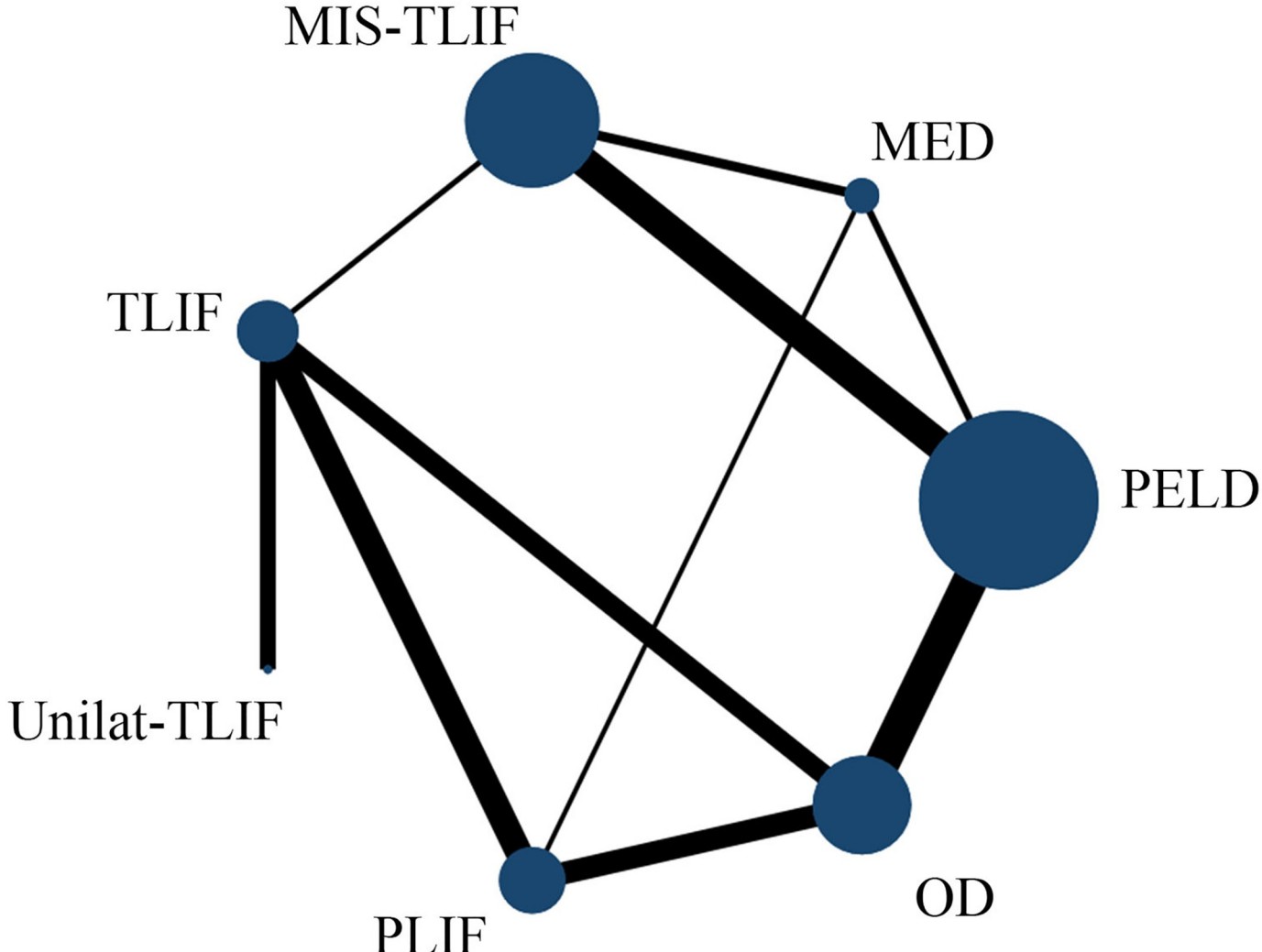

**Fig 2. Network plots of comparison-based NMA.** Each circular node represents a type of intervention. The circle size is proportional to the total number of patients. The width of lines is proportional to the number of studies performing head-to-head comparisons in the same study.

node-splitting analysis (all P > 0.05; S1 Table), It is worth noting that there are no comparisons to assess for inconsistency in hierarchical Bayesian, so we adopted the frequentist method for analysis [43]. These results indicate that the efficacy profile of each approach, except for OD, was the same. The cumulative ranking probabilities (SUCRA) of VAS (back or leg pain) change ranking from high to low is shown in Fig 7B. The probabilities are detailed in the S2 Table.

**Complications.** Fifteen studies (75%) with 1324 patients (85.09%) presented usable results for complications (7 approaches). In the consistency model, MIS-TLIF was significantly lower than PELD and MED in the complication rate (SMD: 0.37(0.14,0.84) and 0.15(0.034,0.68), respectively), TLIF was significantly Lower than MED in the complication rate (SMD:0.14 [0.027,0.70]) (Fig 8A). The results obtained in the consistency model were in good agreement with those obtained in the inconsistency model; node-splitting analysis showed no significant inconsistency (all P > 0.05; Fig 9; S1 Table). The direct and indirect results of different

**Table 2. The characteristics of the studies.**

| Authors and years | Country | methodology | Treatments | | | Demographic | | | Clinical Outcomes |
|---|---|---|---|---|---|---|---|---|---|
| | | | Group 1 | Group 2 | Group 3 | Group 1(Male/Female) | Group 2(Male/Female) | Group 3(Male/Female) | |
| Anqi Wang et al. (2020) | China | Retrospective case series | PELD | MIS-TLIF | — | 24(14/10) mean 49.25±13.95 years | 22(14/8) mean 56.00 ±7.76 years | — | VAS、ODI |
| Yuan Yao et al. (2017) | China | Retrospective case series | PELD | MIS-TLIF | MED | 28(18/10) mean 53.68±17.70 years | 26(13/13) mean 51.62 ±10.04 years | 20(11/9) mean 51.05 ±16.38 years | VAS、ODI、SF-12 |
| Salvatore D'Oria et al. (2023) | Italy | Randomized controlled trial | — | MIS-TLIF | MED | — | 45(25/20) mean 44.3 (range 20–71)years | 45(22/23) mean 46.6 (range 20–73) years | VAS、JOA |
| Gerald Musa et al. (2024) | Russia | Retrospective case series | — | PLIF | MED | — | 34(19/15) mean 49.88 ±9.38 years | 40(18/22) mean 51.28 ±10.02 years | ODI |
| Junlong Wu et al. (2017) | China | Retrospective case series | PELD | MIS-TLIF | — | 47(34/13) mean47.91±14.77 years | 58(42/16) mean46.76 ±12.37 years | — | VAS、ODI、SF-12 |
| Chao Liu et al. (2024) | China | Retrospective case series | PELD | MIS-TLIF | — | 209(110/99) mean 57.2 years | 192(92/100) mean 55.9 years | — | VAS、ODI、JOA |
| Ayman A et al. (2013) | Egypt | Randomized controlled trial | OD | TLIF | PLIF | 15(8/7) mean 41 ±11.10 years | 15(9/6) mean 40.5±9.68 years | 15(8/70 mean 42.7 ±10.40 years | — |
| Ahmed Zaater et al. (2016) | Egypt | Retrospective case series | OD | PLIF | — | 24(N/N) mean 50.2±12.4 years | 15(N/N) mean 52.8±8.6 years | — | JOA |
| Erkin Sonmez et al. (2013) | Turkey | Retrospective case series | — | Unilat-TLIF | TLIF | 10(4/6) mean 47.3 years | 10(5/5) mean 45.6 years | — | VAS、ODI |
| Xianglong Zhuo et al. (2009) | China | Retrospective case series | OD | TLIF | PLIF | 25(17/8) mean 39.5years | 18(13/5) mean 43 years | 22(14/8) mean 41 years | VAS、ODI |
| Yongsheng Hu et al. (2023) | China | Retrospective case series | OD | — | PLIF | 31(17/14) mean 53.65±12.38 years | — | 42(23/19) mean 55.86 ± 13.68 years | VAS、ODI |
| Junhai Lu et al. (2022) | China | Retrospective case series | PELD | — | OD | 56(32 /24) mean 48.79±14.40 years | — | 58(32 /26) mean 48.41 ± 15.14 years | VAS、ODI |
| Hao Xue (2016) | China | Retrospective case series | PELD | — | OD | 18(11/7) mean 47.5±12.0 years | — | 18(10/8) mean 46.0 ±13.5 years | VAS、JOA |
| Xiaogang Hu (2017) | China | Retrospective case series | PELD | — | OD | 53(37/16) mean 45.9±9.7 years | — | 37(26/11) mean 46.4 ±8.3 years | VAS、ODI |
| Tianji Zhang et al. (2017) | China | Retrospective case series | PELD | — | OD | 41(24/17) mean 50.7± 2.0 years | — | 41(23/18) mean 50.4 ±1.9 years | VAS、ODI |
| Jiancheng Su et al. (2016) | China | Retrospective case series | PELD | — | OD | 36(N/N) N | — | 40(N/N) N | VAS、ODI |
| Yinhe Chen et al. (2014) | China | Retrospective case series | OD | TLIF | PLIF | 12(8/4) mean 50.5 years | 26(19/7) mean 49.3 years | 27(19/8) mean 50.7 years | VAS、ODI |
| Guiying Gao et al. (2019) | China | Retrospective case series | — | MIS-TLIF | TLIF | — | 34(23/11) mean 36.71 ± 7.43years | 34(21/13) mean 37.56 ±7.82years | VAS、JOA |
| Liqiang Li et al. (2016) | China | Retrospective case series | — | TLIF | PLIF | — | 26(17/9) mean 43.8 ±12.1years | 25(18/7) mean 44.5 ±12.4years | JOA |
| Bing Pan et al. (2014) | China | Retrospective case series | — | Unilat-TLIF | TLIF | — | 26(17/9) mean 43.8 ±12.1years | 25(18/7) mean 44.5 ±12.4years | VAS、JOA |

Abbreviations: PELD = Percutaneous Endoscopic Lumbar Diskectomy. MED = Microendoscopic Discectomy. MIS-TLIF = Minimally Invasive transforaminal lumbar interbody fusion. TLIF = transforaminal lumbar interbody fusion. Unilat TLIF = Unilateral transforaminal lumbar interbody fusion. PLIF = posterior lumbar interbody fusion. OD = Open discectomy. VAS = visual analog scales. ODI = Oswestry Disability Index. JOA = Japanese Orthopaedic Association Scores. SF-12:12-item Short Form Health Survey.

approaches are shown in Fig 9. These results indicate that the safety profile of each treatment, except for MIS-TLIF and TLIF, was the same. The cumulative ranking probabilities (SUCRA) of complication rate change ranking from high to low is shown in Fig 8B. The probabilities are detailed in the S2 Table.

**Table 3. The methodological quality of the study.**

| Study | A | B | C | D | E | F | G | H | Total scores |
|---|---|---|---|---|---|---|---|---|---|
| Anqi Wang et al. (2020) | 2 | 2 | 2 | 2 | 1 | 1 | 2 | 0 | 12 |
| Yuan Yao et al. (2017) | 2 | 2 | 2 | 2 | 1 | 1 | 2 | 0 | 12 |
| Gerald Musa et al. (2024) | 2 | 2 | 2 | 2 | 1 | 2 | 2 | 0 | 13 |
| Junlong Wu et al. (2017) | 2 | 2 | 2 | 2 | 1 | 1 | 2 | 0 | 12 |
| Chao Liu et al. (2024) | 2 | 2 | 2 | 2 | 1 | 2 | 2 | 2 | 15 |
| Ahmed Zaater et al. (2016) | 2 | 2 | 2 | 2 | 1 | 2 | 2 | 0 | 13 |
| Erkin Sonmez et al. (2013) | 2 | 2 | 2 | 2 | 1 | 2 | 2 | 0 | 13 |
| Xianglong Zhuo et al. (2009) | 2 | 2 | 2 | 2 | 1 | 2 | 2 | 0 | 13 |
| Yongsheng Hu et al. (2023) | 2 | 2 | 2 | 2 | 1 | 1 | 2 | 0 | 12 |
| Junhai Lu et al. (2022) | 2 | 2 | 2 | 2 | 1 | 1 | 2 | 0 | 12 |
| Hao Xue (2016) | 2 | 2 | 2 | 2 | 1 | 2 | 2 | 0 | 13 |
| Xiaogang Hu (2017) | 2 | 2 | 2 | 2 | 1 | 1 | 2 | 0 | 12 |
| Tianji Zhang et al. (2017) | 2 | 2 | 2 | 2 | 1 | 1 | 2 | 0 | 12 |
| Jiancheng Su et al. (2016) | 2 | 2 | 2 | 2 | 1 | 1 | 2 | 0 | 12 |
| Yinhe Chen et al. (2014) | 2 | 2 | 2 | 2 | 1 | 1 | 2 | 0 | 12 |
| Guiying Gao et al. (2019) | 2 | 2 | 2 | 2 | 1 | 1 | 2 | 0 | 12 |
| Liqiang Li et al. (2016) | 2 | 2 | 2 | 2 | 1 | 1 | 2 | 0 | 12 |
| Bing Pan et al. (2014) | 2 | 2 | 2 | 2 | 1 | 1 | 2 | 0 | 12 |

## Recurrence

Nine studies (45%) with 1025 patients (65.87%) presented usable results for complications (6 approaches). In the consistency model, PELD was significantly higher than MIS-TLIF, TLIF and PLIF in the recurrence rate (SMD: 1.3e-17[2.4e-44,0.00016],1.2e-12[2.1e-36,0.34] and 1.4e-12[6.2e-35,0.013], respectively), MED was significantly higher than MIS-TLIF and PLIF in the recurrence rate (SMD: 2.6e-17[5.6e-44,0.0037]and 3.1e-12[1.6e-34,0.022], respectively), OD was significantly higher than MIS-TLIF, TLIF and PLIF in the recurrence rate (SMD:4.6e+16[2.3e+02,3.0e+43], 4.3e+11[2.4,2.5e+35] and 4.1e+11[35,8.7e+33], respectively) (Fig 8A). The results obtained in the consistency model were in good agreement with those obtained in the inconsistency model; node-splitting analysis showed no significant inconsistency (all P > 0.05; Fig 10; S1 Table). The direct and indirect results of different approaches are shown in Fig 10. The cumulative ranking probabilities (SUCRA) of complication rate change ranking from high to low is shown in Fig 8B. The probabilities are detailed in the S2 Table.

## Discussion

Repeat discectomy is recommended as a treatment option for patients with recurrent disc herniation and nerve root disease [2]. This has been supported by extensive research. Fu et al. [44]. pointed out in a Level III retrospective review that for patients with recurrent disc herniation accompanied by sciatica, it is recommended to only perform discectomy without fusion. Chitnavis et al. [45]. reported a group of patients with recurrent disc herniation accompanied by back pain symptoms or signs of lumbar instability. These patients underwent posterior decompression and interbody fusion treatment, proving good results with fusion in these patients with recurrent disc herniations with instability and/or axial low-back pain. Feng Lei et al. [10]. point out in a systematic review that vertebral fusion is superior to repeated intervertebral disc resection in the treatment of recurrent lumbar disc herniation surgery. Although

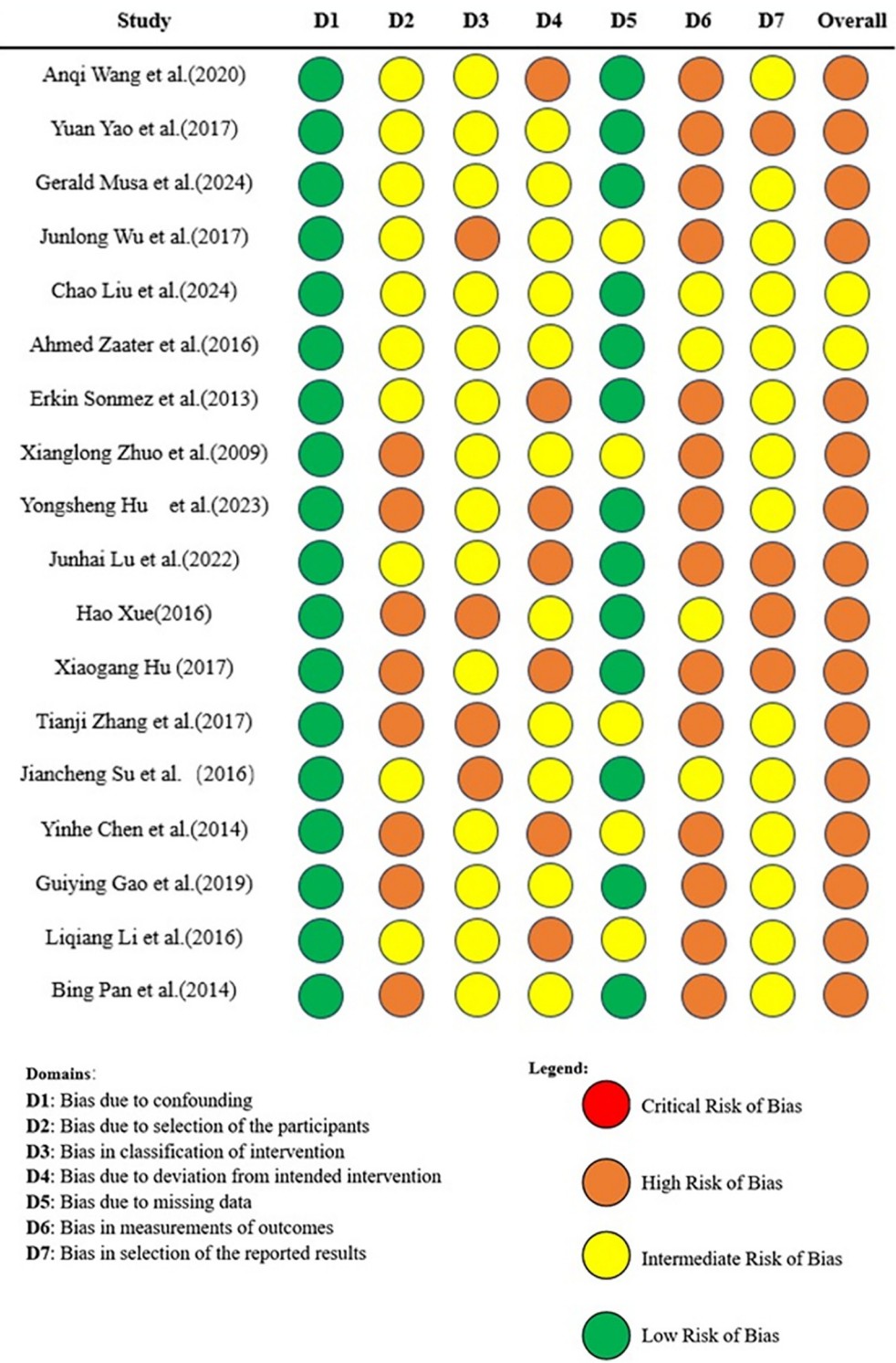

**Fig 3. The risk of bias in non-randomised studies of interventions.**

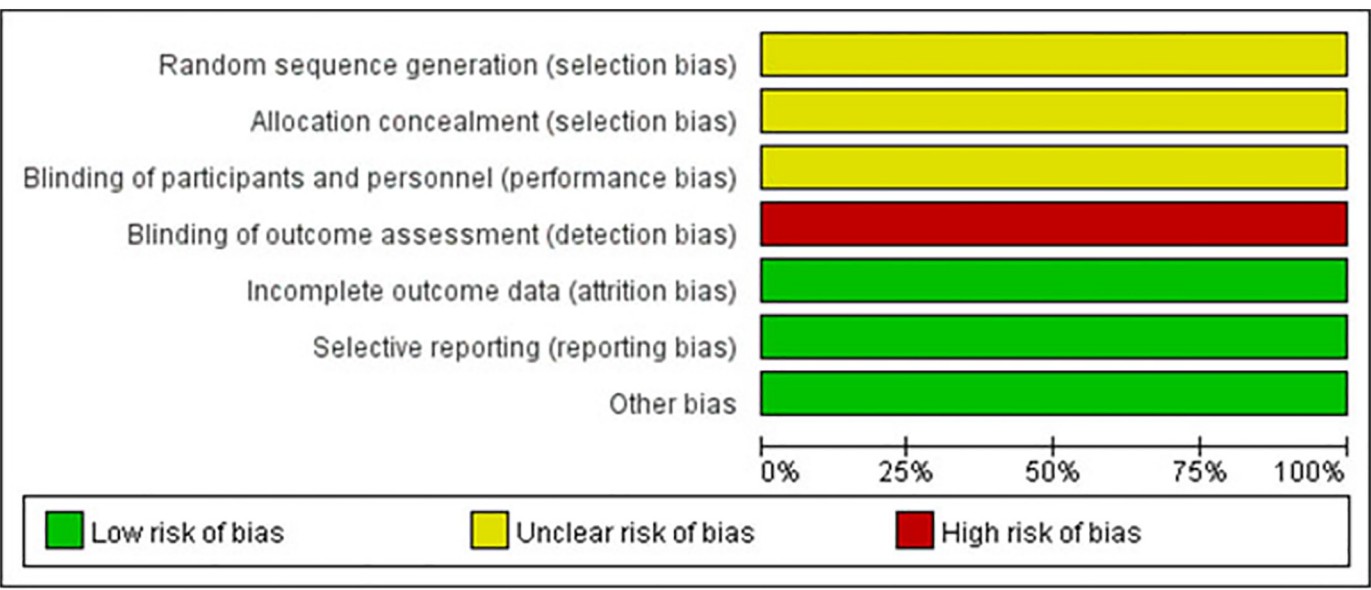

**Fig 4. Risk of bias graph.**

there is still controversy regarding whether vertebral fusion is necessary for the treatment of recurrent lumbar disc herniation surgery, it appears that vertebral fusion can lead to better clinical outcomes. In order to demonstrate whether vertebral fusion is needed in the treatment of recurrent lumbar disc resection surgery and to compare the differences between different surgical methods, we conducted this network meta-analysis.

Our results indicate that in terms of improving postoperative pain, there is no significant difference between non-fusion surgeries (e.g. PELD, MED, and OD) and fusion surgeries (e.g. MIS-TLIF, TLIF, Unilat-TLIF, and PLIF). This is similar to the findings of Feng Lei et al. In terms of improving postoperative function, OD is superior to PELD; however, the study by LI et al. suggests that there is no significant difference between OD and PELD in this aspect. The reason for this discrepancy in results could be that LI et al. only conducted direct comparisons without indirect comparisons, leading to different results. Additionally, in this aspect, OD is also superior to MIS-TLIF. The study examined the complication rates following various surgical interventions, revealing that MIS-TLIF had significantly lower rates compared to PELD and MED. Moreover, TLIF also exhibited significantly lower complication rates than MED, highlighting that vertebral fusion surgery entailed notably reduced complication risks compared to non-fusion surgeries, consistent with previous research findings. Furthermore, concerning postoperative recurrence rates, PELD demonstrated significantly higher recurrence rates than MIS-TLIF, TLIF, and PLIF. Likewise, MED exhibited higher recurrence rates compared to MIS-TLIF and PLIF, while OD also displayed notably higher recurrence rates than MIS-TLIF, TLIF, and PLIF. These results underscore the considerably lower recurrence rates associated with vertebral fusion surgeries relative to non-fusion procedures. Furthermore, a comparative analysis was conducted to assess distinctions among various vertebral fusion procedures. The study revealed that while there was no statistically significant variance in the efficacy of different vertebral fusion techniques regarding the enhancement of postoperative pain relief, functional outcomes, complications, and recurrence rates, the cumulative probability data indicated that MIS-TLIF demonstrated superior outcomes in terms of complications and

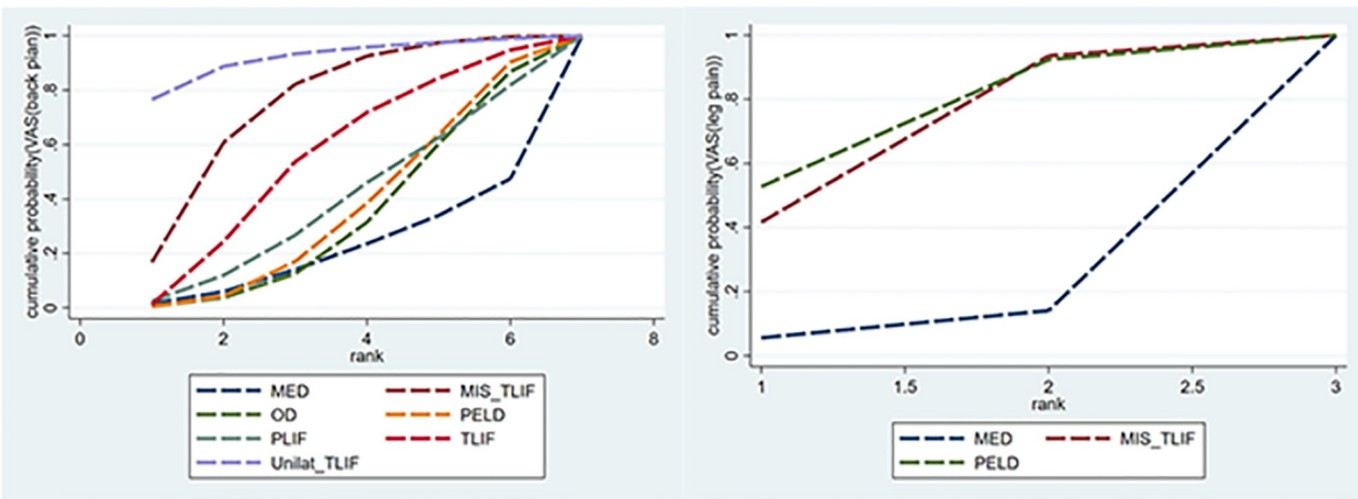

**Fig 5. The core result diagram of VAS (back and leg pain).** (A)VAS back pain and leg pain profile. (B) The cumulative ranking probabilities of VAS back pain and leg pain-based NMA in the consistency model. significant results are in bold.

recurrence rates. In contrast, to open vertebral fusion procedures, MIS-TLIF was associated with reduced operative duration, intraoperative blood loss, and length of hospitalization [46, 47]. In conclusion, it is posited that vertebral fusion surpasses re-discectomy for managing recurrent lumbar disc herniation, and MIS-TLIF presents as a more favorable surgical approach in the context of recurrent lumbar disc herniation treatment.

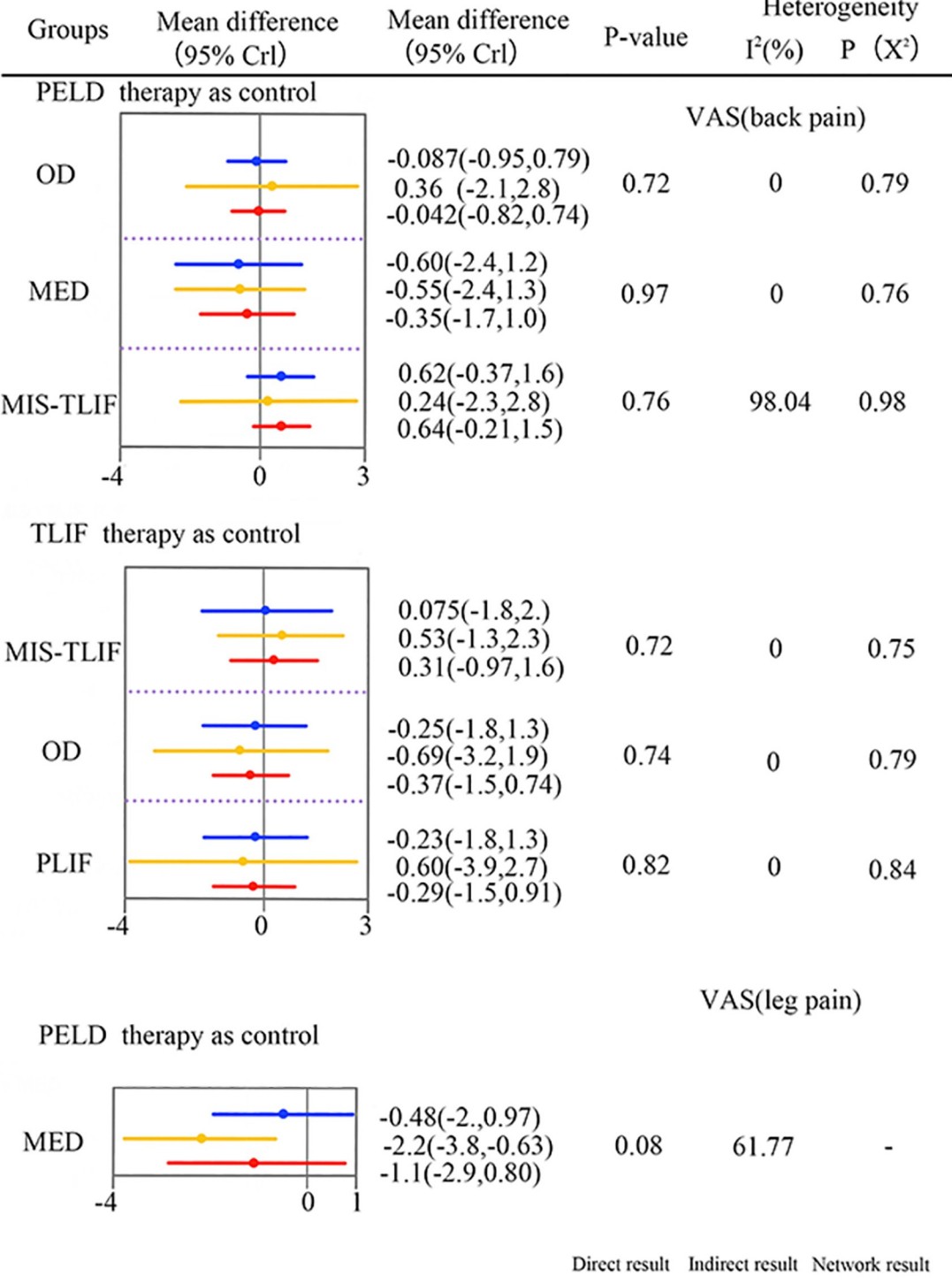

**Fig 6. Forest plots depicting the direct and indirect results of head-to-head comparisons.** *Values in brackets are 95% CrIs.

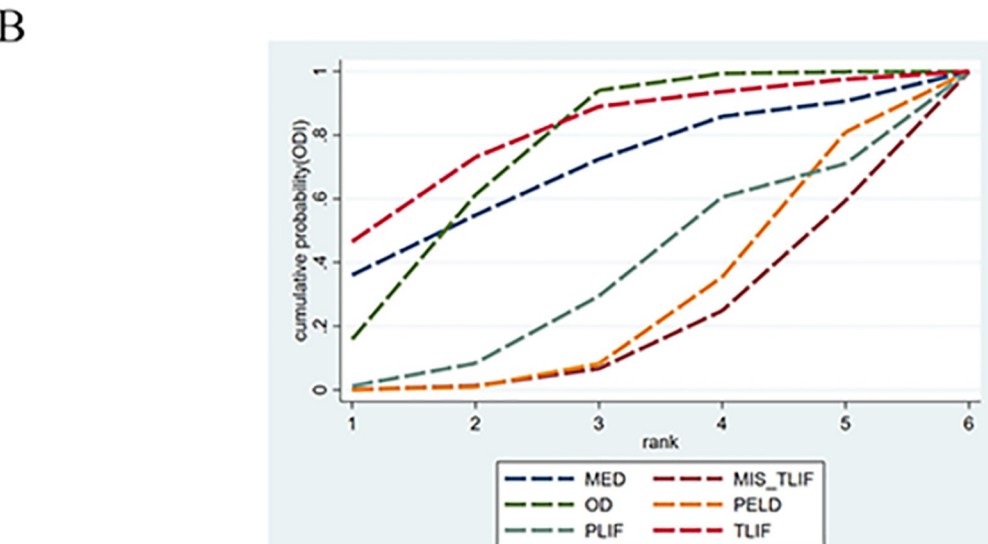

**A**

☐ ODI   ■ Comparison

| PELD | — | — | — | — | — | — |
|---|---|---|---|---|---|---|
| 1.9 (-1.5,5.5) | MED | — | — | — | — | — |
| 0.15 (-0.80,1.0) | 2.0 (-1.4,5.6) | MIS-TLIF | — | — | — | — |
| 2.4 (-0.92,5.7) | 0.46 (-4.3,5.4) | 2.6 (-0.95,6.0) | TLIF | — | — | — |
| — | — | — | — | Unilat -TLIF | — | — |
| 0.41 (-2.3,3.1) | 1.5 (-3,5.9) | 0.56 (-2.3,3.4) | 2.0 (-0.83,4.7) | — | PLIF | — |
| **1.9 (0.21,3.4)** | 0.038 (-3.7,4.0) | **2.0 (0.084,3.8)** | 0.57 (-2.4,3.4) | — | 1.4 (-0.80,3.6) | OD |

**B**

Fig 7. **The core result diagram of ODI.** (A) The profile of ODI. (B) The cumulative ranking probabilities of ODI-based NMA in the consistency model. significant results are in bold.

## Limitations

This study exhibits certain limitations. Primarily, although 20 studies are included and there is barely any obvious heterogeneity and inconsistency among the studies (S3 Table), most of the included studies are retrospective studies, and a notable absence of high-quality randomized controlled trials persists, thereby contributing to a reduction in the overall evidential quality.

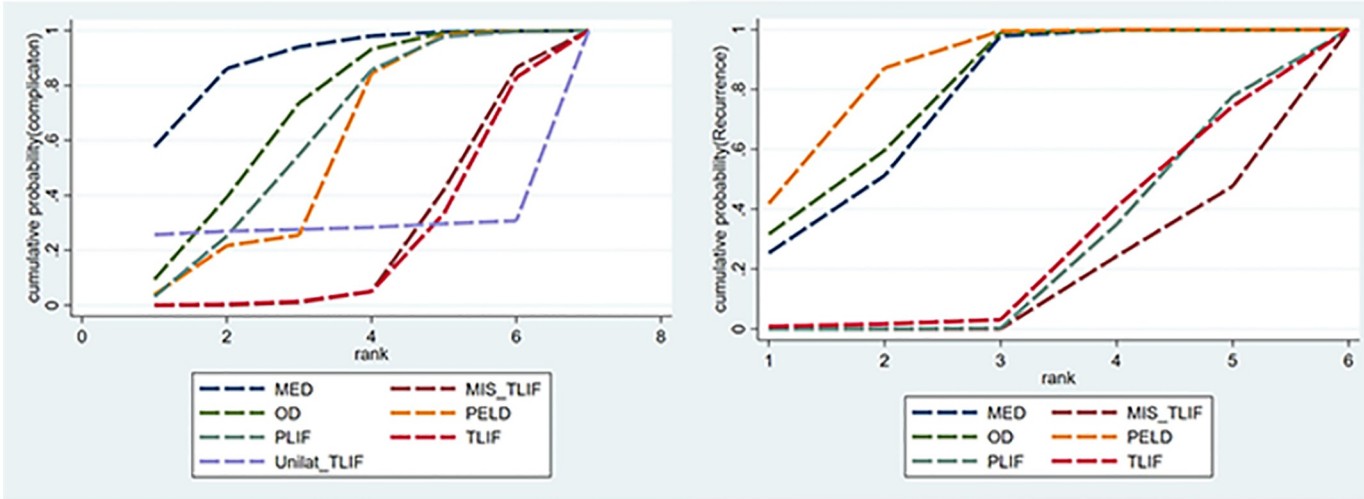

**Fig 8. The core result diagram of complication and recurrence.** (A) The profile of complication and recurrence. (B) The cumulative ranking probabilities of complication and recurrence-based NMA in the consistency model. significant results are in bold.

Secondly, the limitations imposed by the short-term follow-up periods in the included studies constrain the research, with instances where documented complications and recurrences may not be fully depicted. Moreover, while a random effects model was employed for the analysis, it is crucial to acknowledge the significant heterogeneity among certain outcomes, so it is necessary to analyze and interpret the results carefully. Given the existing research constraints, forthcoming studies are encouraged to undertake more high-quality, low-risk randomized

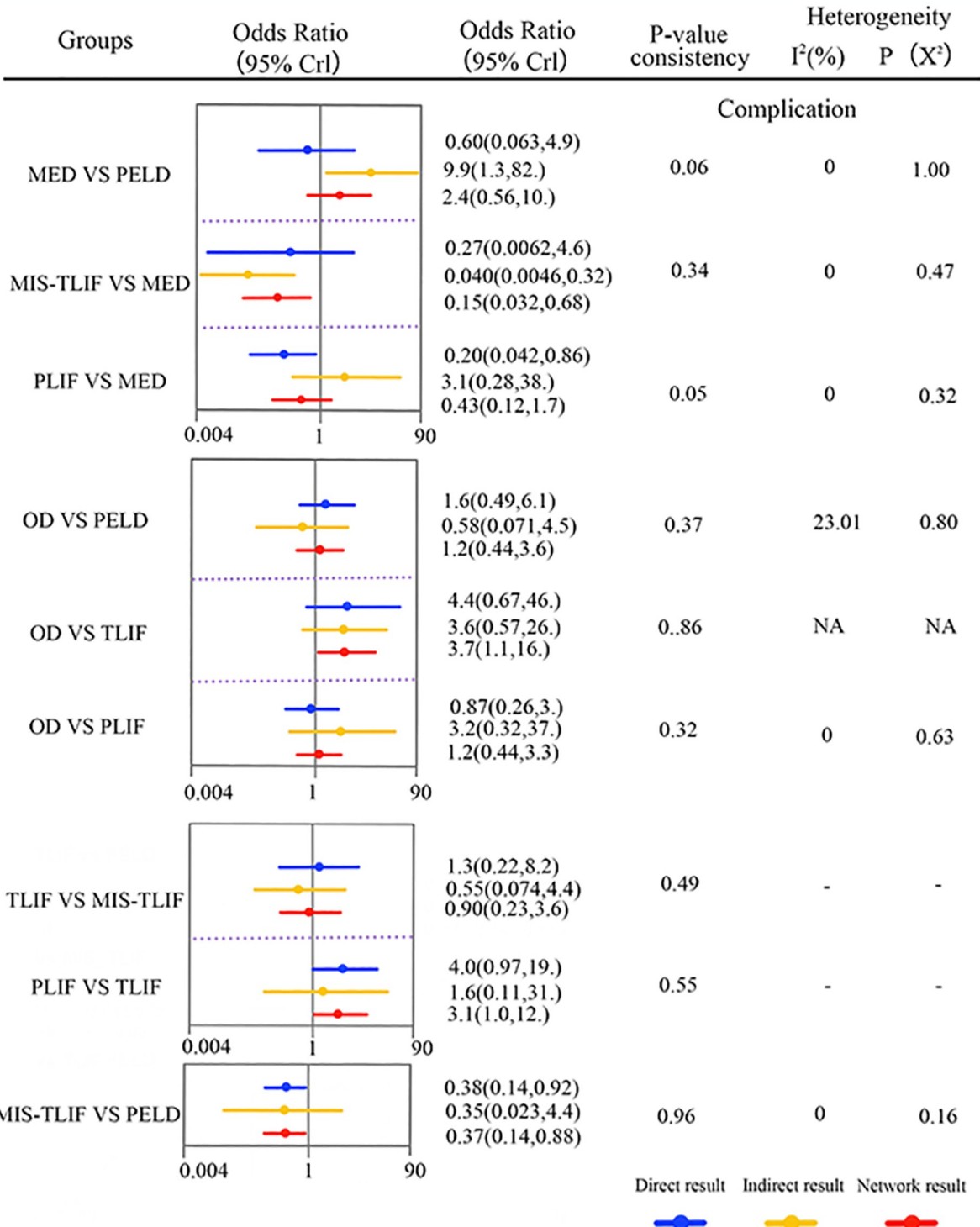

**Fig 9. Forest plots depicting the direct and indirect results of head-to-head comparisons.** *Values in brackets are 95% CrIs.

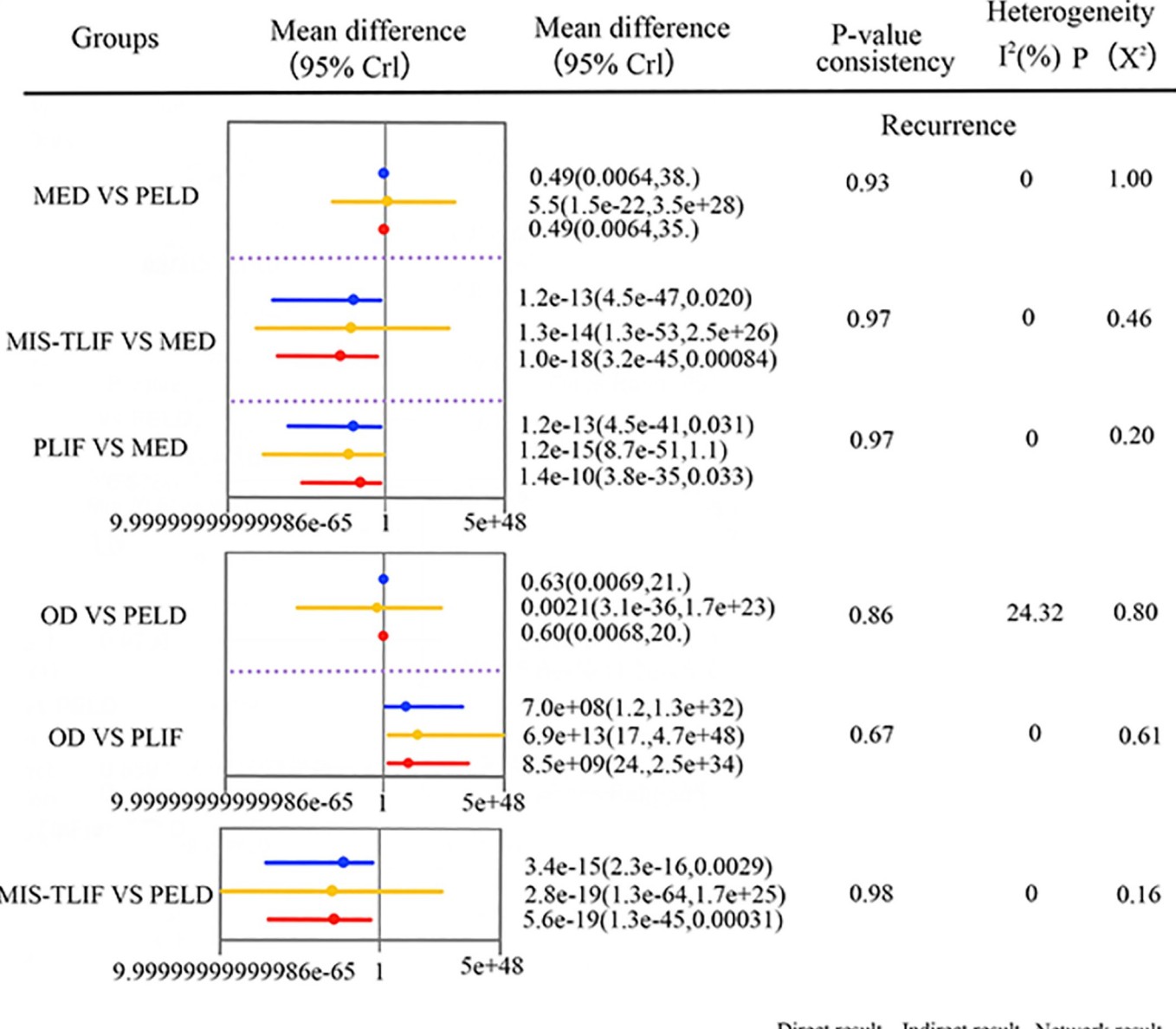

**Fig 10. Forest plots depicting the direct and indirect results of head-to-head comparisons.**

controlled trials, augment sample sizes, and prolong follow-up durations to meticulously evaluate and discern the surgical interventions for recurrent lumbar disc herniation that yield more favorable outcomes.

## Conclusions

In the treatment of recurrent lumbar disc herniation, vertebral fusion surgery is superior to repeat discectomy. At the same time, MIS-TLIF may be a preferable surgical procedure in the treatment of recurrent lumbar disc herniation.

## Supporting information

**S1 Checklist. PRISMA 2020 checklist.**
(DOCX)

**S1 Table. List of node-splitting analyses data.**
(DOCX)

**S2 Table. List of rank possibility data.**
(DOCX)

**S3 Table. List of heterogeneity analysis data.**
(DOCX)

**S4 Table. List of Meta-analysis data of the study.**
(DOCX)

**S5 Table. List of raw data included in the study.**
(DOCX)

**S6 Table. Sifting the list of qualified literature.**
(DOCX)

**S7 Table. GRADE for research result.**
(DOCX)

**S8 Table. Risk of bias in included studies.**
(DOCX)

## Author Contributions

**Conceptualization:** Hang Zhang, Junmao Gao, Qipeng Xie.

**Data curation:** Hang Zhang, Junmao Gao, Qipeng Xie, Mingxin Zhang.

**Formal analysis:** Hang Zhang.

**Investigation:** Hang Zhang, Junmao Gao, Qipeng Xie, Mingxin Zhang.

**Methodology:** Hang Zhang, Junmao Gao, Qipeng Xie, Mingxin Zhang.

**Project administration:** Junmao Gao.

**Software:** Hang Zhang, Junmao Gao, Qipeng Xie, Mingxin Zhang.

**Supervision:** Junmao Gao.

**Writing – original draft:** Hang Zhang.

**Writing – review & editing:** Hang Zhang, Junmao Gao.

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
