## [Decision Letter · Decision Letter 0]

5 Jul 2024

PONE-D-24-22399

Comparison of 7 Surgical Interventions for Recurrent Lumbar Disc Herniation: A Network Meta-analysis and Systematic Review

PLOS ONE

Dear Dr. Gao,

Thank you for submitting your manuscript to PLOS ONE. After careful consideration, we feel that it has merit but does not fully meet PLOS ONE’s publication criteria as it currently stands. Therefore, we invite you to submit a revised version of the manuscript that addresses the points raised during the review process.

We look forward to receiving your revised manuscript.

Kind regards,

Kentaro Yamada, M.D., Ph.D.

Academic Editor

PLOS ONE

Journal Requirements:

https://journals.sagepub.com/doi/10.1177/2192568217745063

https://www.painphysicianjournal.com/current/pdf?article=NzI1OA%3D%3D&journal=136

In your revision ensure you cite all your sources (including your own works), and quote or rephrase any duplicated text outside the methods section. Further consideration is dependent on these concerns being addressed.

5. We note that this manuscript is a systematic review or meta-analysis; our author guidelines therefore require that you use PRISMA guidance to help improve reporting quality of this type of study. Please upload copies of the completed PRISMA checklist as Supporting Information with a file name “PRISMA checklist”.

Reviewers' comments:

Reviewer's Responses to Questions

**Comments to the Author**

1. Is the manuscript technically sound, and do the data support the conclusions?

Reviewer #1: Yes

Reviewer #2: Yes

2. Has the statistical analysis been performed appropriately and rigorously? 

Reviewer #1: Yes

Reviewer #2: Yes

3. Have the authors made all data underlying the findings in their manuscript fully available?

Reviewer #1: Yes

Reviewer #2: Yes

4. Is the manuscript presented in an intelligible fashion and written in standard English?

Reviewer #1: Yes

Reviewer #2: Yes

5. Review Comments to the Author

Reviewer #1: This article is interesting and could be published. The topic is interesting and very useful. The study is serous and with well design. The result is interesting. Also the article had good discussion as well.

Reviewer #2: Thanks for the opportunity to review the important and interesting topic at the core of this paper. It compared 7 surgical interventions for recurrent lumbar disc herniation. However, this article also has some shortcomings. It may not be accepted at present.

1. It was not searched systematically according to the principles of PICO, the MeSH were not clear, and the free words were not comprehensive.

2. The retrieval database is not comprehensive, and the databases that must be retrieved for meta analysis: pubmed, embase, cochrane library; Database to be retrieved as far as possible: web of science, clinicalrials gov，Google Scholar.

3. the registration of the plan had not been completed in PROSPERO.

4. The studies included in the meta-analysis should follow the principle of homogeneity, and the combination of RCT and retrospective study can only increase the heterogeneity. It is recommended that RCT and retrospective study be subgrouped.

6. PLOS authors have the option to publish the peer review history of their article (what does this mean?). If published, this will include your full peer review and any attached files.

Reviewer #1: **Yes: **Chang, Chien-Chun

Reviewer #2: No

---

## [Author Response · Author response to Decision Letter 0]

12 Jul 2024

Dear Editor and reviewers

Re: Manuscript ID: PONE-D-24-22399R1 and Title: Comparison of 7 Surgical Interventions for Recurrent Lumbar Disc Herniation: A Network Meta-analysis and Systematic Review.

 Thank you for your letter and the reviewers' comments concerning our manuscript entitled “Comparison of 7 Surgical Interventions for Recurrent Lumbar Disc Herniation: A Network Meta-analysis and Systematic Review”(ID: PONE-D-24-22399R1). Those comments are all valuable and very helpful for revising and improving our paper, as well as the important guiding significance to our research, We have studied the comments carefully and have made corrections which we hope meet with approval. The main corrections in the paper and the responses to the editor's and reviewer's comments are as follows:

Responds to the editor's comments:

 We are grateful to the reviewers and editorial team for the time and effort they put into evaluating our manuscript. We made detailed revisions to the manuscript as requested. Modify as follows:

1. We have revised the title and figure captions according to the style template of the PLOS ONE manuscript and reflected them in the manuscript.

2. We made changes where there was overlap with the previous publication and this is reflected in the manuscript. (lines 66-70)

3. We have sorted out the raw data needed for this study and uploaded it as supporting information. At the same time, we have attached the description of supporting information files at the end of the manuscript.( line 158 and lines 379-398)

4. We have uploaded the full PRISMA checklist as support information.

Responds to the reviewer's comments:

Reviewer #1: We sincerely thank you for your comments and suggestions, which have greatly facilitated our research.

Reviewer #2:

1. Response to comment:( It was not searched systematically according to the principles of PICO, the MeSH were not clear, and the free words were not comprehensive.)

Response: 

 We think this is an excellent comment. We possess a profound comprehension of the principles of PICO significance in clinical research, and the meticulous exploration of pertinent literature is paramount. Consequently, we have refined the MeSH and free words of this study, conducted additional searches for related studies, and integrated the revised MeSH and free words into the manuscript.(lines 99-109)

2. Response to comment:(The retrieval database is not comprehensive, and the databases that must be retrieved for meta analysis: pubmed, embase, cochrane library; Database to be retrieved as far as possible: web of science, clinicalrials gov,Google Scholar.)

Response: 

 We appreciate your valuable comments. Yes, a comprehensive search of articles from different databases is essential, so we researched relevant articles in different databases again, for example, PubMed, Embase, MEDLINE, Cochrane Library, Web of Science, Google Scholar and China National Knowledge Infrastructure databases, meanwhile, the results of the search are integrated into the manuscript. (lines 97-99 and Fig.1)

3. Response to comment:( the registration of the plan had not been completed in PROSPERO.)

Response: 

 Thank you very much for your comments. the registration of the study had been completed in PROSPERO (Prospero registration number: CRD42024545798). (lines 113-115)

4. Response to comment:( The studies included in the meta-analysis should follow the principle of homogeneity, and the combination of RCT and retrospective study can only increase the heterogeneity. It is recommended that RCT and retrospective study be subgrouped.)

Response:

 We appreciate your valuable comments. Yes, In the meta-analysis, the included study should follow the principle of homogeneity in order to improve the credibility of the evidence, which we deeply agree with. At the same time, the subgroup analysis of RCT and retrospective studies will help to further improve the credibility of the conclusions. We sincerely thank you for your suggestion. In this study, we are also aware of this, but considering that in the clinical study of orthopedic surgery, The RCT is challenging because the choice of surgical methods will be affected by the choice of patients, although some studies have described randomized controlled trials. At the same time, in the included study, the number and sample size of randomized controlled trials are relatively small, which is also one of the limitations of this study. If we analyze the randomized controlled trials in groups, there is a risk of chance. Therefore, after a detailed discussion, we combined randomized controlled trials with a retrospective study in this study. Meanwhile considering the heterogeneity of the study, we adopt the random effect model to analyze and draw a conclusion.

 We deeply recognize the limitations and shortcomings of this study, so in future research, we will include more high-quality, low-risk randomized controlled trials for analysis, increase the sample size, reduce the risk of heterogeneity, and improve the shortcomings of the study. Thank you for your valuable advice again.

 As researchers, we deeply recognize the importance of the review comments. We tried our best to improve the manuscript which will not influence the content and framework of the paper. We appreciate for editors' and reviewers’ warm work earnestly and hope the correction will meet with approval. Once again, thank you very much for your comments and suggestions.

Best regards!

---

## [Decision Letter · Decision Letter 1]

12 Aug 2024

PONE-D-24-22399R1

Comparison of 7 Surgical Interventions for Recurrent Lumbar Disc Herniation: A Network Meta-analysis and Systematic Review

Professor Junmao Gao

PLOS ONE

Dear Professor Gao,

The PLOS ONE has rescinded the decision on manuscript PONE-D-24-22399R1. 

Kind regards,

Melanie Española

Support Staff - Editorial

PLOS ONE

---

## [Author Response · Author response to Decision Letter 1]

12 Oct 2024

Dear Editor 

Re: Manuscript ID: PONE-D-24-22399R1 and Title: Comparison of 7 Surgical Interventions for Recurrent Lumbar Disc Herniation: A Network Meta-analysis and Systematic Review.

 We sincerely apologize for the delay resulting from our researchers' misinterpretation of the submission format and requirements for original research data. We are immensely grateful to the editorial team for their willingness to review and assess our manuscript again. We have reorganized and provided the required research data and reflected it in the manuscript. Modify as follows:

 1. Response to comment:(A numbered table of all studies identified in the literature search, including those that were excluded from the analyses. This should be numbered table with rows from 1-54 and include all publications assessed for eligibility as stated in your PRISMA flowchart.

a. For the 24 excluded studies, the table should include a column which lists the reason(s) for exclusion.

b. If any of the included studies are unpublished, include a link (URL) to the primary source or detailed information about how the content can be accessed.)

 Response:

 We have collated 54 eligible studies and provided DOI numbers, while listing the reasons for excluding 34 studies.(Supplementary table 6)

 2. Response to comment:(A second table of all data extracted from the primary research sources for the systematic review and/or meta-analysis. This table should be a numbered table from 1-20 and include all of the studies included in the systematic review as stated in your PRISMA flow chart. The table must include the following information for each study:

a. A column for the Name of data extractors and a column for the date of data extraction

b. A column for the confirmation that the study was eligible to be included in the review.

c. A column (or columns) that states all data extracted from each study for the reported systematic review and/or meta-analysis that would be needed to replicate your analyses. For example, include a column for clinical outcomes and a column for complications etc.)

 Response:

 We have collated the 20 studies included in this analysis, providing the name of data extractors and the date of data extraction, and collated all the raw data used for the analysis in each study, including visual analogue scale, Oswestry disability index, complications and recurrence(Supplementary table 5). In addition, we also summarized the data used in the Networkmeta-analysis and presented them in the Supplementary table 4.

 3. Response to comment:(A third table showing the completed risk of bias and quality/certainty assessments for each study or outcome. Please ensure this is provided for each domain or parameter assessed. For example, if you used the Cochrane risk-of-bias tool for randomized trials, provide answers to each of the signalling questions for each study. If you used GRADE to assess certainty of evidence, provide judgements about each of the quality of evidence factor. This should be provided for each outcome.)

 Response:

We have employed different risk bias assessment tools tailored to the specific type of each study and provide answers to each of the signaling questions for each study, while summarizing the evaluation results in Supplementary Table 8. Furthermore, we utilized the GRADE methodology to assess the research results, which are presented in Supplementary Table 7.

 At the same time, since the initial submission of the manuscript, we have made revisions in accordance with the recommendations of peer review and editor, as follows:

1. We have refined the MeSH and free words of this study, conducted additional searches for related studies, and integrated the revised MeSH and free words into the manuscript. (lines 99-109)

2. We researched relevant articles in different databases again, for example, PubMed, Embase, MEDLINE, Cochrane Library, Web of Science, Google Scholar and China National Knowledge Infrastructure databases, meanwhile, the results of the search are integrated into the manuscript. (lines 97-99 and Fig.1)

3. The research has been successfully registered in PROSPERO and is reflected in the manuscript(Prospero registration number: CRD42024545798). (lines 113-114)

4. We have revised the title and figure captions according to the style template of the PLOS ONE manuscript and reflected them in the manuscript.

5. We made changes where there was overlap with the previous publication and this is reflected in the manuscript. (lines 66-70)

6.We have uploaded the full PRISMA checklist as support information.

 In addition, we have added a schematic of a Network Meta-Analysis of different interventions to this revision.(Fig 2)

 As researchers, we fully acknowledge the critical importance of submitting valid raw data. We have made every effort to enhance the manuscript. We would like to extend our heartfelt gratitude to the editorial team for their dedicated efforts and sincerely hope that this revision will be favorably received. Once again, thank you for your valuable comments and suggestions.

 Best regards!

---

## [Editor Report · Decision Letter 2]

1 Nov 2024

Comparison of 7 Surgical Interventions for Recurrent Lumbar Disc Herniation: A Network Meta-analysis and Systematic Review

PONE-D-24-22399R2

Dear Dr. Gao,

We’re pleased to inform you that your manuscript has been judged scientifically suitable for publication and will be formally accepted for publication once it meets all outstanding technical requirements.

Kind regards,

Kentaro Yamada, M.D., Ph.D.

Academic Editor

PLOS ONE

---

## [Editor Report · Acceptance letter]

11 Nov 2024

PONE-D-24-22399R2 

PLOS ONE

Dear Dr. Gao, 

I'm pleased to inform you that your manuscript has been deemed suitable for publication in PLOS ONE. Congratulations! Your manuscript is now being handed over to our production team.

Kind regards, 

on behalf of

Dr. Kentaro Yamada 

Academic Editor

PLOS ONE